# Epithelial Regeneration Ability of Crohn’s Disease Assessed Using Patient-Derived Intestinal Organoids

**DOI:** 10.3390/ijms22116013

**Published:** 2021-06-02

**Authors:** Chansu Lee, Sung-Noh Hong, Eun-Ran Kim, Dong-Kyung Chang, Young-Ho Kim

**Affiliations:** 1Samsung Medical Center, Department of Medicine, Sungkyunkwan University School of Medicine, 81 Irwon-ro, Gangnam-gu, Seoul 0635l, Korea; cslhero@gmail.com (C.L.); er.kim@samsung.com (E.-R.K.); do.chang@samsung.com (D.-K.C.); bowelkim@gmail.com (Y.-H.K.); 2Stem Cell & Regenerative Medicine Center, Samsung Medical Center, 81 Irwon-ro, Gangnam-gu, Seoul 06351, Korea

**Keywords:** Crohn’s disease, tumor necrosis factor-alpha, intestinal organoids, epithelial regeneration, wound healing

## Abstract

Little is known about the ability for epithelial regeneration and wound healing in patients with inflammatory bowel diseases. We evaluated the epithelial proliferation and wound healing ability of patients with Crohn’s disease (CD) using patient-derived intestinal organoids. Human intestinal organoids were constructed in a three-dimensional intestinal crypt culture of enteroscopic biopsy samples from controls and CD patients. The organoid-forming efficiency of ileal crypts derived from CD patients was reduced compared with those from control subjects (*p* < 0.001). Long-term cultured organoids (≥6 passages) derived from controls and CD patients showed an indistinguishable microscopic appearance and culturing behavior. Under TNFα-enriched conditions (30 ng/mL), the organoid reconstitution rate and cell viability of CD patient-derived organoids were significantly lower than those of the control organoids (*p* < 0.05 for each). The number of EdU+ cells was significantly lower in TNFα-treated organoids derived from CD patients than in TNFα-treated control organoids (*p* < 0.05). In a wound healing assay, the unhealed area in TNFα-treated CD patient-derived organoids was significantly larger than that of TNFα-treated control organoids (*p* < 0.001). The wound healing ability of CD patient-derived organoids is reduced in TNFα-enriched conditions, due to reduced cell proliferation. Epithelial regeneration ability may be impaired in patients with CD.

## 1. Introduction

The intestinal epithelium, which acts as a frontline defense of the human body, is repetitively injured and continuously regenerated [1]. Epithelial regeneration depends on the self-renewal and proliferation of LGR5+ intestinal stem cells (ISCs), which occur in intestinal crypts [2]. Crohn’s disease (CD) is a chronic relapsing–remitting inflammatory bowel disease (IBD), characterized by cycles of mucosal inflammation and ulceration, followed by regeneration and restoration [3]. Impaired epithelial regeneration can lead to sustained intestinal inflammation and can be accompanied by ulcers and complications, such as fibrosis and fistulas.

Nevertheless, little is known about the ability for epithelial regeneration and wound healing in patients with CD. Clinical trials could not be implemented to evaluate this issue due to ethical reasons and clinical heterogeneity. Classical tumor-derived cell lines and animal model systems have inherent limitations [4]. Immortalized cell lines consist of a homogenous cellular component and evade cellular senescence, due to the presence of certain mutations [5]. Animal models have limitations in replicating human specific biologic processes [6]. With the advent of intestinal epithelium-derived organoids, it is possible to cultivate all epithelial cellular components and re-create the functional crypt–villus architecture [7,8,9]; patient-derived intestinal organoids might be appropriate for studying the regenerative ability of epithelial cells. This study aimed to evaluate the epithelial regenerative ability of CD patient-derived intestinal organoids, compared to that of control subject-derived intestinal organoids.

Inflammation challenges epithelial integrity and barrier function. The intestinal epithelium needs to adapt to a multitude of signals in order to perform the complex process of maintenance and restitution of its barrier function [10]. Tumor necrosis factor-alpha (TNFα) acts as a major pro-inflammatory and tissue damage-promoting effector during the pathogenesis of CD; this is supported by evidence provided by studies involving experimental mouse models and the therapeutic effects of TNFα-neutralizing reagents in IBD treatment [11,12,13]. Recently, mucosal healing has been considered a therapeutic target, as it improves the prognosis of patients with CD [14]. We aimed to understand the TNFα-induced alteration of epithelial regenerative ability in CD patient-derived organoids compared to that of control organoids. 

## 2. Results

Intestinal crypts isolated from duodenal (CD, *n* = 5; controls, *n* = 5), jejunal (CD, *n* = 9; controls, *n* = 13), ileal (CD, *n* = 43; controls, *n* = 15), and colonic biopsy samples (CD, *n* = 12; controls, *n* = 8) from controls (*n* = 34) and patients with CD (*n* = 51) were cultured in Matrigel with maintenance medium and then organoid-forming efficiency was evaluated. Patient characteristics for the enrolled patients are listed in Table 1. In both groups, the organoid-forming efficiency of duodenal crypts was the highest, followed by jejunal, ileal, and colonic crypts. The organoid-forming efficiency of the ileal crypts obtained from patients with CD was significantly lower than that of the ileal crypts obtained from controls on day 3 (53.4% ± 4.0% vs. 39.6% ± 2.4%, *p* = 0.005), day 5 (38.9% ± 3.0% vs. 21.1% ± 1.7%, *p* < 0.001), and day 7 (29.2% ± 2.8% vs. 15.7% ± 1.5%, *p* < 0.001). The organoid-forming efficiency on day 7 of the jejunal crypts obtained from CD patients was numerically lower than that of those obtained from controls (46.8 ± 2.6% vs. 38.1% ± 3.8%, Figure 1).

Organoids grown from intestinal crypts were sub-cultured in maintenance medium and the control organoids became uniformly spheroid (>90% of total organoids) after 2–4 passages. However, organoids derived from patients with CD had both enteroid and spheroid forms in the early passages and became uniformly spheroid after 4–6 passages. After six passages, the organoids derived from the controls and patients with CD exhibited consistent spheroid features and culturing behaviors (Figure 2).

The shapes of the organoids cultured in the maintenance medium were similar regardless of their location; however, those cultured in differentiation medium tended to have different shapes, depending on their origin. The murine enteroids tended to have more budding structures compared to human enteroids. Ileal organoids typically exhibited budding, whereas jejunal organoids formed thick-walled structures (Figure 3). Previous studies have noted that the cytotoxicity of intestinal organoids occurred in a concentration-dependent manner in response to TNFα [15,16]. To address the appropriate concentration of TNFα and the interval of administration to assess the epithelial regenerative ability, control organoids were tested for changes in the expression of the ISC and progenitor marker after TNFα treatment and organoid survival at the various concentrations of TNFα. The organoid viability, measured using MTT and the enteroid/spheroid ratio, decreased significantly with a gradual increase in the TNFα concentration. At TNFα concentrations of ≤10 ng/mL in the differentiation medium, changes in cell viability and morphology were negligible; however, the changes observed at TNFα concentrations of ≥30 ng/mL were notable (Appendix A). The expression of *LGR5* (active ISC), *BMI1* (reserve ISC), *HES1* (absorptive progenitor), and *ATOH1* (secretory progenitor) increased within 24 h of TNFα treatment and then decreased (Figure 4).

Based on these results, organoids derived from controls and CD patients, cultured over the long-term (≥6 passages), were treated with 30 ng/mL TNFα every 24 h for 10 days (Figure 5A). The epithelial regenerative ability of intestinal organoids was evaluated using organoid reconstitution, MTT, EdU, and wound healing assays. The organoid reconstitution rate of TNFα-treated organoids was significantly lower than that of TNFα-free organoids (jejunal organoids: 68.9% ± 12.4% vs. 47.4% ± 13.6%, *p* < 0.001; ileal organoids: 55.2% ± 12.3% vs. 32.8% ± 10.1%, *p* < 0.001). There was no significant difference in the organoid reconstitution rate between TNFα-free controls and CD patient-derived organoids; however, the organoid reconstitution rate of TNFα-treated CD patient-derived organoids was significantly lower than that of TNFα-treated control organoids (jejunal organoids: 55.5% ± 11.5% vs. 39.3% ± 10.6%, *p* = 0.011; ileal organoids: 40.2% ± 6.9% vs. 25.3% ± 6.6 %, *p* = 0.027; Figure 5B,C).

The organoid viability was assessed using MTT; the results showed that the formazan absorbance values of viable TNFα-treated organoids were significantly lower than those of TNFα-free organoids (OD of jejunal organoids: 0.96 ± 0.16 vs. 0.71 ± 0.12, *p* < 0.001; OD of ileal organoids: 0.83 ± 0.16 vs. 0.56 ± 0.16, *p* < 0.001). In the TNFα-enriched condition, the viable cells of jejunal and ileal CD patient-derived organoids were significantly decreased compared with those of the control organoids (OD of jejunal organoids: 0.79 ± 0.10 vs. 0.62 ± 0.07, *p* = 0.135; OD of ileal organoids: 0.66 ± 0.13 vs. 0.45 ± 0.10, *p* = 0.019; Figure 6).

Two hours after EdU administration, EdU+ cells were confirmed in the buds, which represented the intestinal crypt (Appendix A). The number of EdU+ cells was higher in TNFα-free organoids than in TNFα-treated organoids (91.9 ± 32.6 vs. 32.3 ± 22.2, *p* < 0.001). Although there was no significant difference in the number of EdU+ cells between the control and CD patient-derived organoids in the steady-state (82.7 ± 33.1 vs. 102.4 ± 30.5, *p* = 0.653), the number of EdU+ cells were significantly lower in TNFα-treated CD patient-derived organoids than in TNFα-treated control organoids (49.0 ± 19.1 vs. 15.6 ± 7.7, *p* = 0.001, Figure 7).

The wound healing assay showed that the unhealed wound area in TNFα-treated CD patient-derived organoids was significantly larger than that in control organoids at 8 h (50.5% ± 10.5% vs. 84.7% ± 12.3%, *p* < 0.001), 16 h (11.0% ± 4.8% vs. 64.3% ± 14.0%, *p* < 0.001), and 24 h after insert removal (1.7% ± 1.5% vs. 37.0% ± 8.5%, *p* < 0.001). There was no significant difference in wound healing between the organoids derived from controls and CD patients. The unhealed wound area at 24 h was not significantly different between the TNFα-free and TNFα-treated control organoids (1.7% ± 1.5% vs. 16.0% ± 4.6%, *p* = 0.257); however, there was a significant difference between the TNFα-free and TNFα-treated CD patient-derived organoids (2.3% ± 2.5% vs. 37.0% ± 8.5%, *p* < 0.001). In addition, the unhealed wound area at 24 h of TNFα-treated CD patient-derived organoids was significantly larger than that in TNFα-treated control organoids at 24 h after insert removal (16.0% ± 4.6% vs. 37.0% ± 8.5%, *p* = 0.044). The wound-healing ability of TNFα-treated CD patient-derived organoids was significantly lower than that of TNFα-free control and CD patient-derived organoids and TNFα-treated control organoids (Figure 8).

RNA-seq was performed on endoscopic biopsy tissue samples from controls and patients with CD and organoids derived from these samples. The clustering heatmap and principal component analysis identified that the gene expression profile was separated between the endoscopic biopsy tissue samples and the organoids derived from these samples. Furthermore, among the gene expression profile of organoids, there was a clear distinction between the TNFα-free and -treated organoids (Figure 9A,B). The epithelial lineage-specific gene expression was evaluated in the tissue samples and organoids derived from these samples (Figure 9C). In endoscopic biopsy tissue samples, the expression of genes associated with the intestinal microbiota (NOD2, DEFA5, DEFA6, PLA2G2A, MUC2, and NARP6) and differentiated cells (enterocytes (ECs): SI, APOC3, ALPI, and APOA1; goblet cells (GCs): MUC2; Paneth cells (PCs): WNT3, ARG2, DLL1, and DLL4; and enteroendocrine cells (EECs): CCK, CHGA, CHGB, and NEUROG3)) was increased in comparison to organoids. The expression of EC markers such as VIL1, KLF5, and KRT5, and GC markers such as TFF3 and MUC13 was upregulated in biopsy tissue samples and TNFα-free control organoids (day 6 and 9 organoids). In the TNFα-treated control organoids, the expression of ISC markers, such as LGR5, OLFM4, and TNFRSF19, was increased compared with those of TNFα-treated CD patient-derived organoids. The alteration of ISC properties can affect cell proliferation and wound healing ability.

## 3. Discussion

Rapid restoration of epithelial defects following injuries or physiologic damage is indispensable for maintaining gut homeostasis [10]. CD is characterized by chronic and transmural inflammation that can occur along the entire GI tract, but primarily occurs in the small intestine. Nevertheless, the ability for epithelial regeneration and wound healing in patients with CD has not been evaluated, especially by the location in the gastrointestinal tract. To our knowledge, this study is the first to evaluate organoid-forming efficiency using patient-derived organoids in patients with CD according to the location in the GI tract.

Several previous observations suggested that epithelial regeneration and wound healing capacity was impaired in patients with CD. Patients with CD have a higher rate of postoperative complication, in particular anastomotic complications leading to intra-abdominal sepsis, than patients without inflammatory status [17]. Exposure to ionizing radiation causes cell death of rapidly proliferating cells, leading to acute or chronic gastrointestinal (GI) toxicity in a dose- and time-dependent manner [18]. Patients with CD presented an increased risk of GI toxicity following exposure to therapeutic doses of ionizing radiation [19]. The mutation of CD susceptibility genes, such as NOD2 and ATG16L1, has been associated with ISC dysfunction, leading to impaired epithelial regeneration and wound healing [20,21]. IBD patient-derived colonoids showed decreased organoid size and amount of budding, increased cell death and luminal debris, and inverted polarization [22]. However, the epithelial regeneration ability of CD patients was not directly evaluated in these previous studies, whereas our study evaluated epithelial regeneration capability in a detailed and direct manner.

This study identified that the organoid-formation ability of CD patient-derived ileal crypts was significantly impaired; this was correlated with the fact that the ileum is the most frequently affected area in CD [23]. These ileal crypts were endoscopically obtained from the uninflamed intestine; however, the effect of pre-existing microscopic inflammation on organoid formation cannot be completely excluded. Long-term cultured CD patient-derived organoids were passaged under non-inflammatory culture conditions and showed no significant differences in morphology and culturing behaviors compared to control organoids, which can mimic the mucosal healing in patients with IBD. Mucosal healing is considered to be an ideal therapeutic target for the long-term remission of IBD [3]. However, while patients achieve mucosal healing, several triggers such as inflammatory cytokines and nonsteroidal anti-inflammatory drugs can cause mucosal damage and inflammation. In this study, TNFα was used as a trigger to mimic the inflammatory milieu. TNFα reduced the cell viability and organoid reconstitution ability of intestinal organoids in a dose-dependent manner. We identified TNF 30 ng/mL as an appropriate concentration for the experiment, because the surviving cells showed morphological changes and were sufficient for additional experiments. Previous studies have also suggested that 30 ng/mL of TNFα may induce cytotoxicity and distinctive cell responses in intestinal organoids [15,16]. In a culture condition with 30 ng/mL of TNFα, both jejunal and ileal organoids derived from CD patients showed reduced organoid reconstitution ability, cell viability, cell proliferation, and wound healing capability compared with control organoids.

Previous studies have shown that wound healing is accomplished through epithelial restitution, ISC proliferation, and differentiation, although these wound healing processes overlap [17,18,19]. This study evaluated the epithelial restitution using wound healing assays and achieving organoid cell proliferation using organoid reconstitution, MTT, and EdU assay. Post-injury wound healing is regulated by a wide range of regulatory factors, including cytokines, growth factors, adhesion molecules, and phospholipids [10,20]. Inflammatory processes have especially been thought to interfere with epithelial cell migration and proliferation, and thus modulate intestinal epithelial healing [21]. However, in this study, RNA-seq identified that the expression of active ISC markers such as LGR5, OLFM4, and TNFRSF19 was decreased in CD patient-derived organoids compared with the control organoids under a TNFα-enriched condition. Under TNFα-enriched culture conditions, the ISC population may be expanded for reparative proliferation to replace lost cells with TNFα-induced cytotoxicity. However, alterations in ISC properties in organoids derived from CD patients in response to TNFα may affect ISC proliferation, and subsequently the wound healing ability can be impaired. Previous single-cell RNA-seq also identified alterations in ISC properties in CD patient-derived small intestinal organoids [24], which supports our RNA-seq results. Based on our results, inflammatory processes can particularly interfere with ISC proliferation, and therefore modulate intestinal epithelial wound healing.

A limitation of our study is that the organoid culture system does not reflect the effects of intestinal microbiota, dietary components, and the mucosal immune system. The intestinal microbiota and dietary components contribute to the fine-tuning of ISC survival and differentiation [25]. In addition, only ileal crypts of CD showed a significant reduction in organoid-formation efficiency compared with those of controls. As enteroscopy was usually performed for the diagnosis and treatment of CD via the per-anal route, ileal sampling was relatively convenient and capable of obtaining a sufficient sample size but duodenal and jejunal sampling were not. The lack of significance of the duodenal, jejunal, and colonic organoid reconstitution rates might be attributed to the small sample size.

The aim of this study was to evaluate whether the epithelial regeneration ability is impaired in patients with CD. The organoid-forming efficiency of the ileal crypts in patients with CD was reduced compared with those of the control subjects. When TNFα was used as a trigger to mimic the inflammatory milieu, the epithelial regeneration and wound healing ability was decreased more noticeably in CD patient-derived organoids compared to in control organoids. Our findings suggest the epithelial regeneration ability was impaired in patients with CD. Clinical trials are unable to settle this issue due to ethical reasons and clinical heterogeneity, while our results indicated that the epithelial regenerative ability is impaired in patients with CD, especially in TNFα-enriched conditions. Impaired epithelial regeneration results in defective mucosal integrity and sustained intestinal inflammation, leading to ulcers, fibrosis, and fistulas, which are the main indications for surgery in patients with CD. Mucosal healing can be regarded as epithelial regeneration after injury of the intestinal lining. Mucosal healing is considered a therapeutic target of IBD [26]. To improve mucosal healing in patients with CD, additional treatment options to promote epithelial regeneration and wound healing ability should be explored and developed.

## 4. Materials and Methods

### 4.1. Sampling

To establish intestinal organoids, we used human intestinal tissue from controls and patients with CD using biopsy forceps during single-balloon enteroscopy at the Samsung Medical Center, Seoul Korea, between November 2016 and December 2018. At least four biopsy samples were obtained from mucosal tissues in the jejunum (100–150 cm distal of the ligamentum of Treitz), ileum (50–100 cm proximal to the ileocecal valve), and colon (transverse colon). Patients with CD were diagnosed according to the guidelines [27]. In patients with CD, biopsies were performed at least 5 cm away from the ulcers. All samples were acquired with informed consent. This study was approved by the institutional ethical committee of the Samsung Medical Center (IRB No. 2016-02-022, approval date: 10/12/2016). 

### 4.2. Crypt Isolation from the Biopsy Specimens 

Endoscopic biopsy samples were incubated in PBS with 10 mM ethylenediaminetetraacetic acid (Thermo Fischer Scientific, San Jose, CA, USA) and 1 mM dithiothreitol (Thermo Fischer Scientific) at 4 °C for 30 min, and then vortexed for 30–120 s. The supernatant containing crypts was filtered through 70-μm cell strainers (Corning, Bedford, MA, USA) and suspended in basal medium (advanced Dulbecco’s modified Eagle’s medium (DMEM)/F12 (Thermo Fischer Scientific) supplemented with antibiotic–antimycotic solution (Thermo Fischer Scientific), HEPES (Thermo Fischer Scientific), GlutaMAX (Thermo Fischer Scientific), N2 (Thermo Fischer Scientific), B27 (Thermo Fischer Scientific), and *N*-acetylcysteine (Sigma-Aldrich, St. Louis, MO, USA). 

### 4.3. Three-Dimensional (3D) Intestinal Crypt Culture 

Isolated crypts were resuspended in Matrigel (Corning) and plated in 48-well culture plates (Corning). After incubation at 37 °C for 15 min, 250 μL of maintenance medium (50% Wnt3a-conditioned medium (ATCC#CRL-2647, Manassas, VA, USA) and 50% of 2× basal medium, supplemented with recombinant human EGF (Sigma-Aldrich), recombinant human noggin (R&D Systems, Minneapolis, MN, USA), recombinant human R-spondin1 (PeproTech, Cranbury, NJ, USA), nicotinamide (Sigma-Aldrich), p160ROCK inhibitor (Selleck Chemicals, Houston, TX, USA), p38 MAP kinase inhibitor (SB202190, Sigma-Aldrich), and Prostaglandin E2 (PGE2, Cayman Chemical, Ann Arbor, MI, USA), were added to the wells. A GSK3 inhibitor (Stemgent, Cambridge, MA, USA) was added to the medium during the first 2 days. 

Depending on their shape, organoids can be classified into spheroids and enteroids. Spheroids are defined as round- or oval-shaped organoids with a thin wall composed of a single layer of undifferentiated cells. Enteroids are defined by the presence of visually sharp borders (buddings) along their basolateral (anti-luminal) side or irregularly thickened walls, which consist of all components of epithelial cells [28] (Figure 10).

### 4.4. Organoid-Forming Efficiency

One hundred crypts obtained from control subjects and CD patients were plated in 25 µL of Matrigel in maintenance medium. The organoid-forming efficiency was calculated as the percentage of viable organoids per 100 intestinal crypts.

### 4.5. Organoid Subculture, Maintenance, and Differentiation

After 7 days of the culture process, the organoids were mechanically disrupted and suspended in cell dissociation buffer (Thermo Fischer Scientific). Single cells and small cell clusters were resuspended in Matrigel and plated in 48-well culture plates (Corning). The medium was changed every 2 days, and the organoids were passaged at a ratio of 1:2–1:4 on day 7. After 6 passages, most organoids in the maintenance medium formed uniform spheroids and could be subcultured stably for a long time in vitro.

To recreate the physiological parameters of the intestinal epithelium, the spheroids were cultured in a differentiation medium (maintenance medium without Wnt3A conditional medium, SB202190, nicotinamide, and PGE2). The differentiation medium was changed every 2 days and enteroids were grown for 7–12 days.

### 4.6. Organoid Reconstitution Assay

After more than six passages, the organoids were mechanically disrupted and suspended in the cell dissociation buffer. Single cells were resuspended in Matrigel and plated in 48-well culture plates. The intestinal organoids were cultured in maintenance medium for 2 days to obtain a stable number of organoids before inducing organoid differentiation; then, the maintenance medium was changed to a differentiation medium every 2 days. Different concentrations of human recombinant TNFα (R&D Systems) were added to the culture medium every 24 h. The organoid reconstitution rate was calculated as the percentage of the number of viable organoids on day 9–10 proportional to the number of viable cells on day 2.

### 4.7. 3-(4,5-Dimethylthiazolyl-2)-2,5-diphenyltetrazolium Bromide (MTT) Assay

Ten microliters of MTT (Sigma-Aldrich) was added to each well of the culture plates and incubated for 3 h until purple precipitates were visible. After 100 μL of detergent reagent was added, the organoids were incubated at room temperature in the dark for 2 h, prior to recording the absorbance at 570 nm. The optical density (OD) of organoid-containing wells is expressed in a ratio based on the values of TNFα-free control organoid-containing wells.

### 4.8. 5-Ethynyl-2′-Deoxyuridine (EdU) Assay

Two hundred micrograms of EdU (Abcam, Cambridge, UK) was added to the culture medium 2 h before fixation with cold 4% paraformaldehyde (Biosesang. Seongnam-si, South Korea). Incorporation of EdU into DNA was detected using a Click-iT™ EdU Alexa Fluor^®^ 488 imaging kit (Thermo Fischer Scientific). The number of EdU (+) cells was measured in 10 organoids (size > 100 μm) selected from the TNFα-free and -treated enteroids derived from the controls and CD patients.

### 4.9. Wound Healing Assay

3D cultured organoids were digested into single cells using TrypLE Express (Thermo Fischer Scientific) and 5 ×10^4^ cells were seeded into 24-well plates containing CytoSelect™ 24-Well Wound Healing Assay inserts (Cell Biolabs, San Diego, CA, USA) [29]. Organoid monolayers were cultured in maintenance medium until confluence was reached. The inserts were then carefully removed to produce 0.9 mm diameter wounds, and fresh differentiating medium was added to each well. The area of the unhealed wound was measured in three different areas. The area of the unhealed wound is expressed in %, based on the observed area of TNFα-free control organoids.

### 4.10. Real-Time Quantitative Reverse Transcription Polymerase Chain Reaction (qPCR)

One-step qPCR was performed using One Step PrimeScript™ III RT-qPCR Mix (Takara, Kusatsu, Japan) with primers as follows: LGR5 primer (Forward: 5′-aactttggcattgtggaagg-3′, Reverse: 5′-acacattgggggtaggaaca-3′), BMI1 primer (Forward: 5′-cgtgtattgttcgttacctgga-3′, Reverse: 5′-ttcagtagtggtctggtcttgt-3′), ATOH1 primer (Forward: 5′-cagctgcgcaatgttatccc-3′, Reverse: 5′-ttgtagcagctcggacaagg-3′), and HES1 primer (Forward: 5′-tttcctcattcccaacgggg-3′, Reverse: 5′-ctggaaggtgacactgcgtt-3′).

### 4.11. RNA Sequencing

RNA sequencing was conducted on endoscopic biopsy tissue samples from controls and patients with CD and organoids derived from these samples. Total RNA was extracted from intestinal organoids using an RNeasy Mini Kit (QIAGEN, Hilden, Germany). RNA was isolated from endoscopic biopsy tissue samples from controls (*n* = 2) and patients with CD (*n* = 2, non-inflamed area, during infliximab maintenance treatment). In addition, organoid culture was conducted with intestinal crypts derived from these specimens. Organoids were sub-cultured for least 6 passages. RNA was isolated from the organoids on day 3, day 6, and day 9, and organoids were treated with TNFα 30 ng/mL on day 3, day 6, and day 9. RNA sequencing was performed using total RNA samples with >10 μg of RNA and an integrity number >8. The libraries were constructed for whole-transcriptome sequencing using the TruSeq RNA Sample Preparation Kit v2 (Illumina, San Diego, CA, USA) and sequenced using the 100-bp paired-end mode of the TruSeq Rapid PE Cluster Kit and the TruSeq Rapid SBS Kit (Illumina).

Reads from files in the FASTQ format were mapped to the hg19 human reference genome using HISAT 2.2.0, with default parameters (https://daehwankimlab.github.io/hisat2/, accessed on 18 May 2021). Raw read counts mapped to genes were measured using the BAM format file in HTSeq version 0.12.3 (https://htseq.readthedocs.io/, accessed on 18 May 2021), to quantify transcript abundance. The coding genes were selected, and raw read counts were normalized to the trimmed mean of M-values. Differential expression analysis of RNA-seq experiments was conducted with edgeR (version 3.28.1). Unsupervised hierarchical clustering analysis with the Euclidean distance and complete linkage algorithm was used to create a heatmap with the associated dendrogram.

### 4.12. Availability of Data and Materials

The datasets presented in this study can be found in GEO (accession number = GSE173294, https://www.ncbi.nlm.nih.gov/geo/query/acc.cgi?acc=GSE173294, accessed on 18 May 2021).

## Figures and Tables

**Figure 1 ijms-22-06013-f001:**
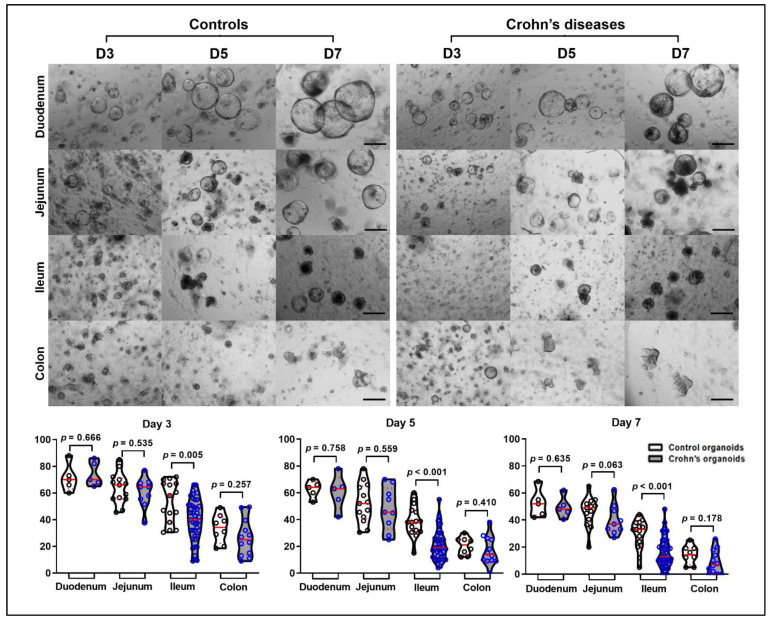
Organoid-forming efficiency. Intestinal crypts were isolated from the duodenum (*n* = 5), jejunum (*n* = 9), ileum (*n* = 43), and colon (*n* = 12) of patients with CD and the duodenum (*n* = 5), jejunum (*n* = 13), ileum (*n* = 15), and colon (*n* = 8) of controls. Differences in organoid-forming efficiency between controls and CD patients based on the location of GI tract were assessed by *t*-test; Scale bar = 200 μm.

**Figure 2 ijms-22-06013-f002:**
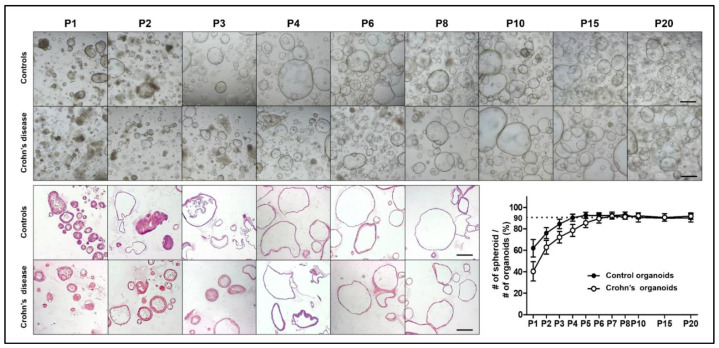
Long-term culture of control and CD patient-derived organoids. Bright field and H&E staining images of control and CD patient-derived organoids according to the passages. After six passages, the morphology of control and CD patient-derived organoids became identical. Scale bar = 200 μm.

**Figure 3 ijms-22-06013-f003:**
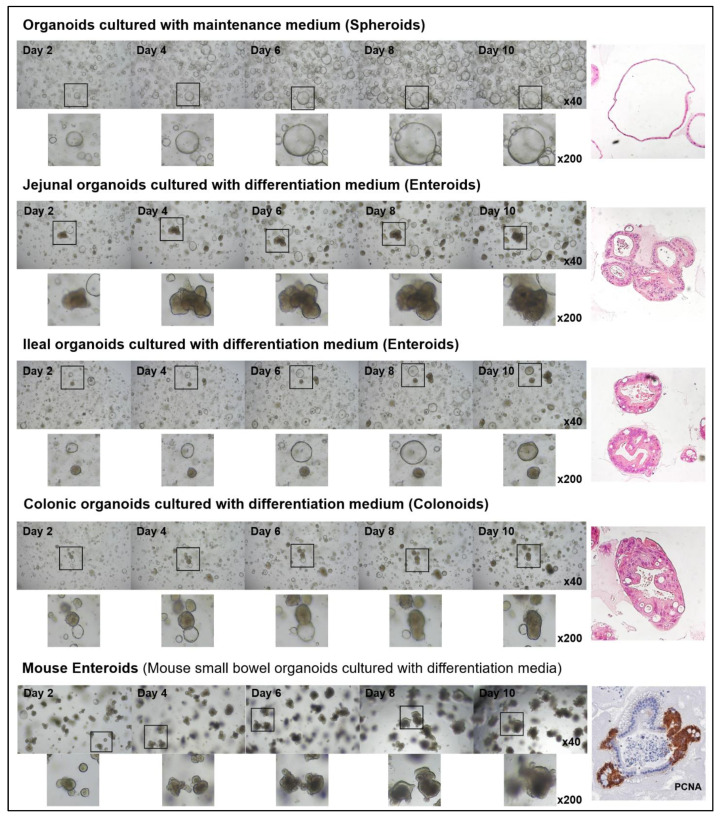
Microscopic appearance of jejunal, ileal, and colonic organoids cultured in differentiation medium. The control organoids cultured in the maintenance medium formed spheroids, while those cultured in the differentiation medium formed enteroids. Jejunal organoids had budding structures, while the ileal and colonial organoids formed thick-walled structures.

**Figure 4 ijms-22-06013-f004:**
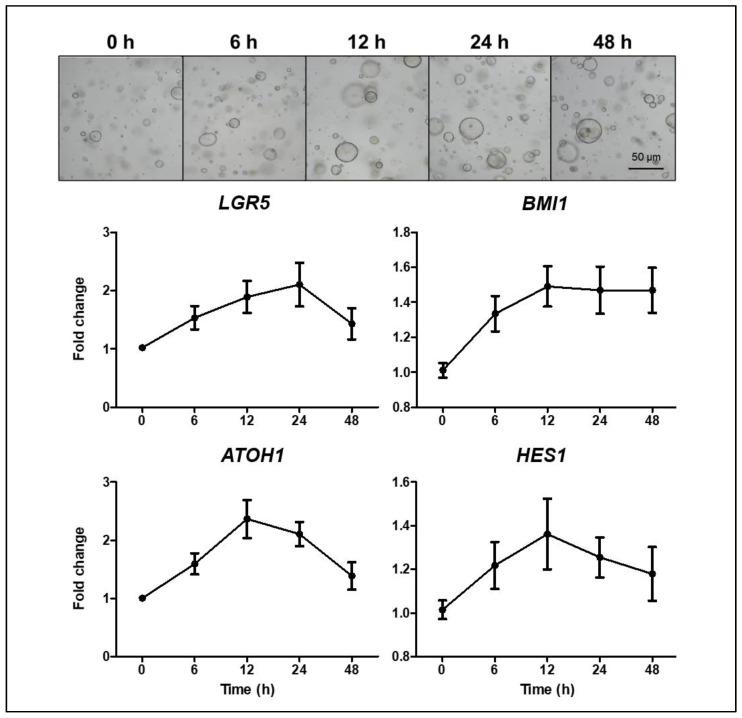
Changes in the expression of *LGR5*, *BMI1*, *ATHO1*, and *HES1* in control organoids at 6, 12, 24, and 48 h after treatment with TNFα (30 ng/mL). Quantitative reverse transcription polymerase chain reaction was performed with control organoids (*n* = 3) in triplicate.

**Figure 5 ijms-22-06013-f005:**
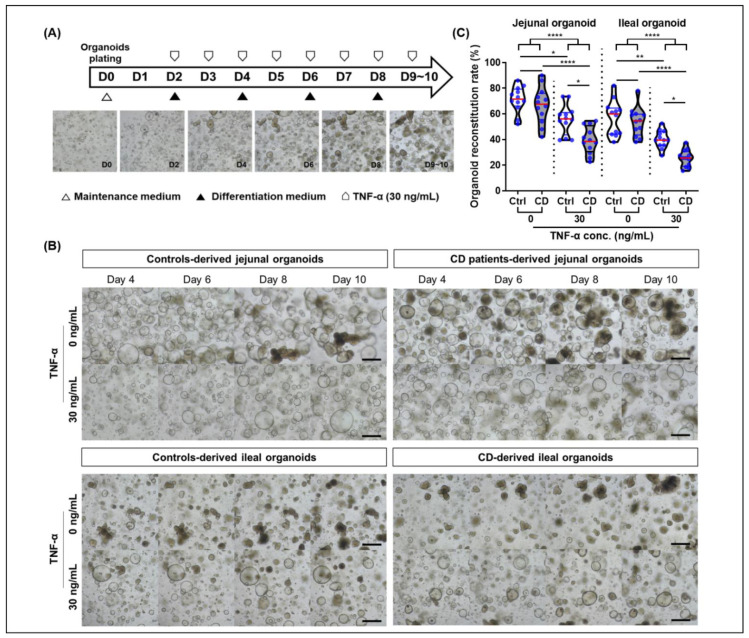
Organoid reconstitution assay in a tumor necrosis factor-alpha (TNFα) enriched condition. (**A**) Study flow diagram. (**B**) Organoid reconstitution assay of jejunal and ileal organoids derived from controls and CD patients (*n* = 5 each) in a TNFα-enriched condition. Assays were conducted in triplicate. (**C**) Organoid reconstitution rate. Reconstituted organoid number is expressed as a percentage value, based on values of TNFα-free jejunal organoids derived from controls. Differences were evaluated using ANOVA with Bonferroni’s multiple comparison test; * *p* < 0.05, ** *p* < 0.01 and **** *p* < 0.0001.

**Figure 6 ijms-22-06013-f006:**
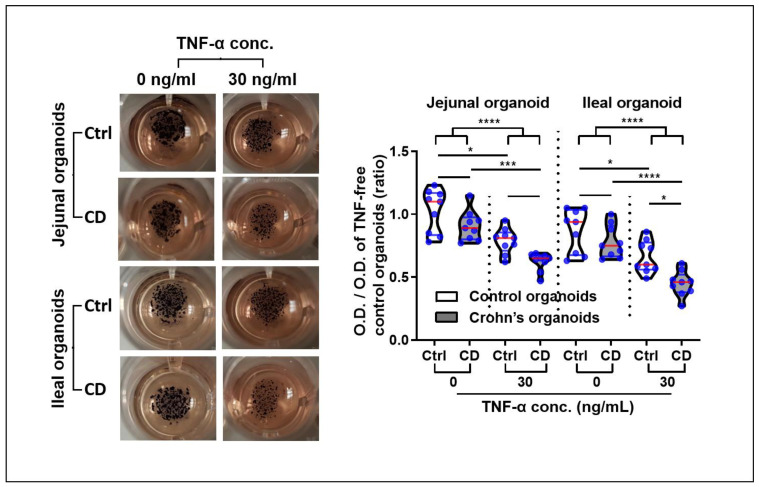
MTT assay for organoid viability in a TNFα-enriched condition. MTT assays were performed in triplicate using jejunal and ileal organoids derived from controls and CD patients (*n* = 3 each). The optical density (OD) of organoid-containing wells is expressed in a ratio based on the values of the TNFα-free control organoid-containing wells. Differences were evaluated using ANOVA with Bonferroni’s multiple comparison test; * *p* < 0.05, *** *p* < 0.001 and **** *p* < 0.0001.

**Figure 7 ijms-22-06013-f007:**
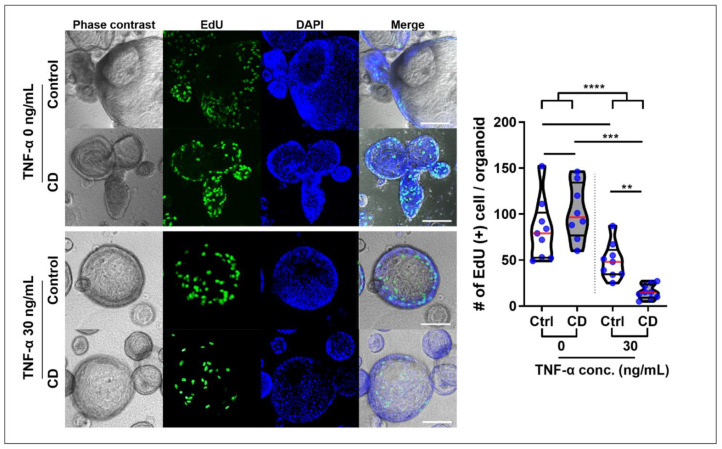
EdU assay for organoid proliferation in a TNFα-enriched condition EdU+ cell number was measured 2 h after EdU administration to 10 organoids from TNFα-treated and -free control and CD patient-derived organoids (*n* = 3 each). Differences were evaluated using ANOVA and Bonferroni’s multiple comparison test; ** *p* < 0.01, *** *p* < 0.001 and **** *p* < 0.0001.

**Figure 8 ijms-22-06013-f008:**
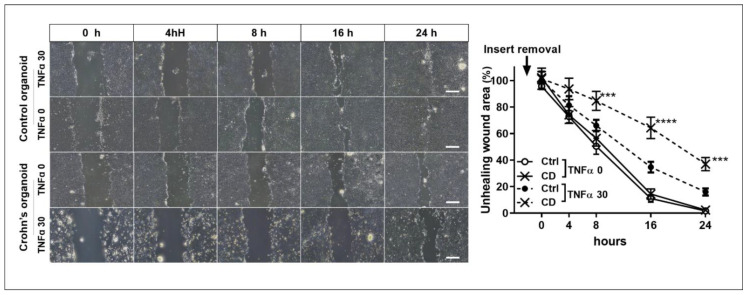
Wound healing assay. Non-healing wound areas in three different areas selected from TNFα-free and -treated organoids derived from controls and CD patients were measured (*n* = 3 each). The area of the unhealed wound is expressed in %, based on the observed area of TNFα-free control organoids. Differences were evaluated using two-way ANOVA test with Bonferroni’s multiple comparison test; *** *p* < 0.001 and **** *p* < 0.0001. Scale bar = 4 mm.

**Figure 9 ijms-22-06013-f009:**
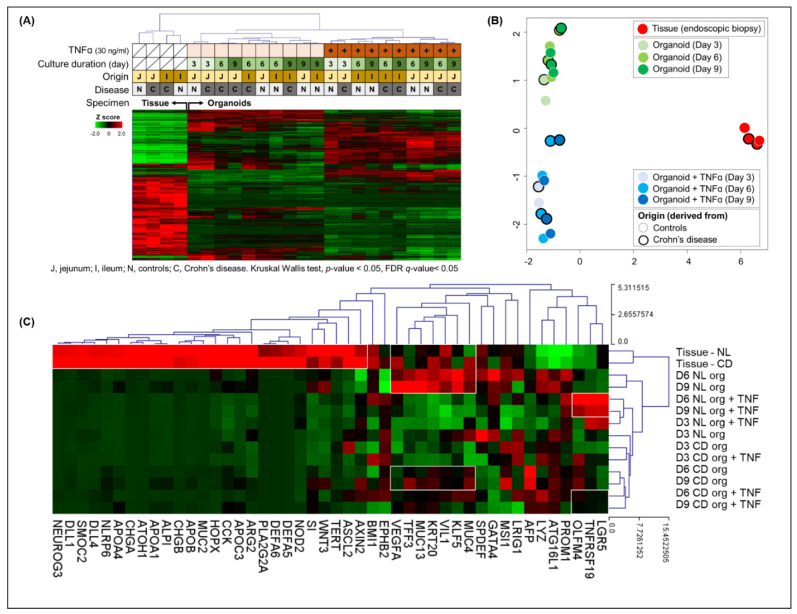
Gene expression profile of the endoscopic biopsy tissue samples from controls and patients with CD and organoids derived from these samples. (**A**) Clustering heatmap and (**B**) principal component analysis. The gene expression profile was clearly separated between the endoscopic biopsy tissue samples and the organoids derived from these samples. Among the gene expression profiles of organoids, a clear distinction was noticed between TNFα-free and -treated organoids. (**C**) Epithelial lineage-specific gene expression. The expression of differentiated cell markers was increased in endoscopic tissue biopsy samples. However, the expression of enterocyte markers, such as VIL1, KLF5, and KRT5, and goblet cell markers, such as TFF3 and MUC13, was increased in the biopsy tissue samples, as well as the TNFα-free organoids. In the TNFα-treated control organoids, the expression of ISC markers, such as LGR5, OLFM4, and TNFRSF19, was increased compared with those of TNFα-treated CD patient-derived organoids.

**Figure 10 ijms-22-06013-f010:**
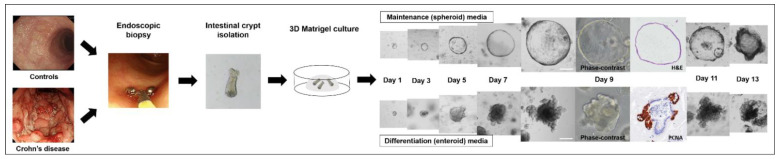
Flow diagram of the patient-derived intestinal organoid model. Scale bar = 100 μm.

**Table 1 ijms-22-06013-t001:** Characteristics of the enrolled patients, and the number of attempts at constructing organoids.

Total Number of Enrolled Patients	85	Number of Attempts to Construct Organoids	110 *
Patients with CD, n	51	Number of sub-culture (≥passage 3) enabled organoids	61
Age, years (mean ± S.D)	36.2 ± 13.4	CD-patient derived organoids (n of attempts/n of subculture)	69/37
Male, n	43	Duodenal organoids (n of attempts/n of subculture)	5/4
Montreal classification, n		Mucosal healing or no duodenal involvement	5/4
Age at diagnosis (A1/A2/A3)	0/42/9	Active ulcer	0/0
Location (L1/L2/L3)	1/34/16	Jejunal organoids (n of attempts/n of subculture)	9/8
Behavior (B1/B2/B3)	20/27/4	Mucosal healing or no jejunal involvement	4/4
IBD Medication at sampling, n		Active ulcer	5/4
None	6	Ileal organoids (n of attempts/n of subculture)	43/23
Immunomodulator	25	Mucosal healing or no ileal involvement	8/5
Anti-TNFα	25	Active ulcer	35/18
Sampling modality, n		Colonic organoids (n of attempts/n of subculture)	12/2
Colonoscopy	11	Mucosal healing or no colonic involvement	4/1
Single-balloon enteroscopy	40	Active ulcer	8/1
Controls, n	34	Control organoids (n of attempts/n of subculture)	41/24
Age, years (mean ± S.D)	50.6 ± 17.9	Duodenal organoids (n of attempts/n of subculture)	5/4
Male sex, n	24	Jejunal organoids (n of attempts/n of subculture)	13/11
		Ileal organoids (n of attempts/n of subculture)	15/8
		Colonic organoids (n of attempts/n of subculture)	8/1

* Twenty-one patients having samples from two different sites and two patients having samples from three different sites.

## Data Availability

The datasets presented in this study can be found in GEO (accession number = GSE173294, https://www.ncbi.nlm.nih.gov/geo/query/acc.cgi?acc=GSE173294, accessed on 18 May 2021).

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
