# Peer review of "Epithelial Regeneration Ability of Crohn’s Disease Assessed Using Patient-Derived Intestinal Organoids"

_ijms, 2021, doi:10.3390/ijms22116013_

Round 1
Reviewer 1 Report
The authors have performed an experiment of Crohn's disease affected organoids of the ileum to healthy controls and compared healing when under the influence of TND alpha. They found that healing appeared reduced in the Crohn's patient organoids compared to controls. This is an interesting concept that provides thought provoking findings for readers regarding the effects the pathological process of Crohn's disease may have on the intestine. I have the following suggestions: The methods section should go before the results and clear aims of the study prior to initating the experiments should be described. Wound healing needs to be defined prior to initiating experiments and the aims, hypotheses, statistical methods and ethical considerations should be outlined. Minor points: - Lines 43 and 44 "Animal models are difficult to replicate human-specific biological processes" consider changing to " There are limitations in replicating human specific biologic processes" - Line 59 "We should attempt" consider changing to "We aimed" - Line 253 "cells were not dying too much" consider revising wordingAuthor Response
We sincerely thank you for the opportunity to revise our manuscript. We have revised the manuscript based on the reviewers’ helpful comments and recommendations. We have done our best to address to these comments thoroughly, and our responses are as follows:
-----------------------------------------
- The methods section should go before the results and clear aims of the study prior to initating the experiments should be described. Wound healing needs to be defined prior to initiating experiments and the aims, hypotheses, statistical methods and ethical considerations should be outlined.
(Answer) Thank you for your comment. We changed the order of the method.
- Minor points:
2.1. Lines 43 and 44 "Animal models are difficult to replicate human-specific biological processes" consider changing to " There are limitations in replicating human specific biologic processes"
(Answer) Thank you for kind comment. We changed “Animal models are difficult to replicate human-specific biological processes" to " Animal models have limitations in replicating human specific biologic processes”
2.2. Line 59 "We should attempt" consider changing to "We aimed"
(Answer) Thank you for kind comment. We changed the words as you suggested.
2.3. Line 253 "cells were not dying too much" consider revising wording
(Answer) Thank you for kind comment. We changed " the cells were not dying too much and presented morphological changes" to “the surviving cells showed morphological changes and were enough for additional experiments”
Reviewer 2 Report
In Chansu Lee et al., the authors performed the epithelial regeneration ability of Crohn's disease assessed using patient-derived intestinal organoids.
Good interesting work, well-conducted research, good discussion section.
Minor concerns:
In conclusion, the authors state „our results indicated that the epithelial regenerative ability is impaired in patients with CD, especially in TNFα-enriched conditions. Our findings suggest the underlying unexplained mechanism for the CD that impaired epithelial regeneration results in defective mucosal integrity and sustained intestinal inflammation, leading to ulcers, fibrosis, and fistulas, which are the main indications for surgery in patients with CD”, which is nothing new in terms of Crohn's disease.
So please justify what was the purpose of this research? Did the research somehow contribute to explaining the pathogenesis and treatment of Crohn's disease?
Author Response
We sincerely thank you for the opportunity to revise our manuscript. We have revised the manuscript based on the reviewers’ helpful comments and recommendations. We have done our best to address to these comments thoroughly, and our responses are as follows
In conclusion, the authors state „our results indicated that the epithelial regenerative ability is impaired in patients with CD, especially in TNFα-enriched conditions. Our findings suggest the underlying unexplained mechanism for the CD that impaired epithelial regeneration results in defective mucosal integrity and sustained intestinal inflammation, leading to ulcers, fibrosis, and fistulas, which are the main indications for surgery in patients with CD”, which is nothing new in terms of Crohn's disease.
So please justify what was the purpose of this research? Did the research somehow contribute to explaining the pathogenesis and treatment of Crohn's disease?
(Answer)
Thank you for kind comment. We changed the conclusion part as following:
The aim of this study was to evaluate whether the epithelial regeneration ability is impaired in patients with CD. The organoid-forming efficiency of the ileal crypts in patients with CD was reduced compared with those of control subjects. When TNFα was used as a trigger to mimic the inflammatory milieu, the epithelial regeneration and wound healing ability was decreased more remarkably in CD patient-derived organoids, compared to that in control organoids. Our findings suggest the epithelial regeneration ability was impaired in patients with CD. The clinical trials are disabled to settle this issue due to ethical reasons and clinical heterogeneity, our results indicated that the epithelial regenerative ability is impaired in patients with CD, especially in TNFα-enriched conditions. impaired epithelial regeneration results in defective mucosal integrity and sustained intestinal inflammation, leading to ulcers, fibrosis, and fistulas, which are the main indications for surgery in patients with CD. Mucosal healing can be regarded as epithelial regeneration after injury of the intestinal lining. Mucosal healing is considered a therapeutic target of IBD